# Morphological and chemotaxonomical characterization of some species of the genus *Euphorbia* L. in Jazan region, KSA

Yehia Hazzazi[1], Mari Sumayli[1], A. El-Shabasy[1*], Abeer Al-Andal[2], Uzma Hanif[3,4], Asmaa Khamis[5], Emad Abada[1,6], Sameh R. Elgogary[7], Taha A. I. El-Bassossy[8], Ahmed A. M. Abdelgawad[7,8]

1 Department of Biology, College of Science, Jazan University, Jazan, Saudi Arabia, 2 Department of Biology, College of Science, King Khalid University, Abha, Saudi Arabia, 3 Department of Botany, Government College University, Lahore Pakistan, 4 Center for Functional Ecology: Science for People & Planet, Marine Resources, Conservation and Technology-Marine Algae Lab, Department of Life Sciences, University of Coimbra, Coimbra, Portugal, 5 Department of Botany, Faculty of Science, Fayoum University, Fayoum, Egypt, 6 Environment and Nature Research Centre, Jazan University, Jazan, Saudi Arabia, 7 Department of Physical sciences, Chemistry division, College of Science, Jazan University, Jazan, Saudi Arabia, 8 Medicinal and Aromatic Plants Department, Desert Research Center, Cairo, Egypt

* ael-shabasy@jazanu.edu.sa

## Abstract

This study decisively evaluates the classification of four species of *Euphorbia*: *Euphorbia ammak*, *Euphorbia fractiflexa*, *Euphorbia granulata*, and *Euphorbia hirta*, collected from diverse habitats in Jazan region (Saudi Arabia). Our objective is to clearly define the interrelationships among these species by utilizing both traditional morphological analyses and cutting-edge chemotaxonomical methods. The morphological analysis examines various aspects of plant life, encompassing qualitative and quantitative parameters. Phytochemical analysis effectively measures total phenolics, alkaloids, flavonoids, saponins, and tannins. High-Performance Liquid Chromatography (HPLC) is employed to capture the phenolic patterns, thereby validating our chemotaxonomic approach. The HPLC analysis unequivocally identifies eleven phenolic and seven flavonoid compounds in the methanol extracts of the four *Euphorbia* taxa. The data collected from the studied Operational Taxonomic Units (OTUs) were meticulously organized into a binary matrix, establishing a similarity matrix and phenogram cluster. Duncan's range test robustly determines the significance of interrelations among the species. The results demonstrate that all examined plant species are rich in phenolic constituents, albeit in varying concentrations. Notably, *Euphorbia granulata* stands out as the most transitional species among them. Taxonomically, our phenogram, based on taxonomic characteristics, reveals two distinct groups: the first group, at a distance of 1.90, includes *Euphorbia ammak* and *Euphorbia fractiflexa*, while the second group, at a distance of 1.52, encompasses the remaining two species. This study strongly recommends considering both adaptation and habitat type when conducting chemotaxonomic analyses of plant species.

**Data availability statement:** All datasets were achieved, used, and analyzed in the current study. They are included in this manuscript.

**Funding:** The author(s) received no specific funding for this work.

**Competing interests:** The authors have declared that no competing interests exist.

## 1. Introduction

Phytochemicals are valuable guides for the most recent sciences, including pharmacology, applied biotechnology, and chemotaxonomy. The use of secondary metabolites from biosynthetic pathways, such as tannins, flavonoids, phenolics, saponins, and alkaloids, is considered the proper way for the determination of relations among plant species [1]. It can be used to solve taxonomic problems and detect the main gaps among transition-related plant species. Moreover, it can predict the medicinal benefits of some other unstudied taxa by referring to their position in chemotaxonomy. Phytochemical profiling of plant species for a specific genus is feasible for different researchers to obtain rationale studies based on their different specializations [2,3].

Several studies reported the pharmacological activities [4], including antiproliferative [5], cytotoxic [6], antimicrobial [7], antipyretic-analgesic [8], inhibition of HIV-1 viral infection [9], ethnomedicinal usage [10,11], and the structural diversity of isolated phytoconstituents [12,13] of *Euphorbia* species.

Euphorbiaceae Juss. (Spurge family) is one of the largest families of flowering plants, conspicuous throughout the tropics, and composed of over 300 genera and 8000 species [14].

*Euphorbia* L. is the largest genus in the family Euphorbiaceae with about 2000 species ranging from herbs to trees [15]. The *Euphorbia* genus consists of species of great economic importance, which makes it a complex genus with great research potential.

There are different phytochemical metabolites, especially in the aerial part of *Euphorbia* species, like flavonoids, terpenoids, alkaloids, tannins, polyphenols, and steroids. They have a great role in different aspects of human demands in pharmacological and industrial activities. Most of them are used as drugs and commercial value products due to their active potential constituents [16].

Phytochemical screening of *E. prolifera* and *E. spinidens* showed different terpenoids and flavonoids acquiring neuroprotective and immunomodulatory activities, respectively [17,18]. Baniadam *et al.* (2014) [19] studied the cytotoxicity of the methane extract of *E. macrostegia* against tumor investigations like MCF-7 and MDA-MB48 due to presence of Cycloartane Triterpenoids. Findings of *E. hirta* extracts indicated antimicrobial activities against different pathogenic Gram (+) and (-) strains [20] besides antioxidant properties that have defense mechanisms originating from different habitats [21]. Moreover, *E. cornigera* and *E. fischeriana* expressed anthelmintic and antiviral activities, respectively [22]. *E. milii* was a candidate to fight different human diseases due to the presence of tannins and flavonoids [23].

The most robust and modern separated phytochemical technique is HPLC. It can separate a mixture of organic chemical compounds with high quality and measure the quantity precisely. It can analyze natural products with standard purification and rapid processing. It provides phytochemical data to researchers who can describe the chemical properties and identify the classes of active fractions [24].

According to the plenty of phytochemicals, Euphorbiaceae can live and adapt in many different zones: tropical, subtropical, and temperate ones [25].

*Euphorbia* spp. have unique morphological structures like pseudodanthial special type of inflorescence called cyathium, colorful tending and sub-tending bracts, cyathial nectar glands provided with petaloid appendages, as well as succulent xerophytic stems protruding with horny thorns. Architecture leaf characterizations include many modified details on marginal configurations, venation types, and leaf shape patterns. The distinct adapted morphology of the genus *Euphorbia* distinguishes its entire species and causes significant taxonomic segregation that needs more research and studies. *Euphorbia* spp. are highly rich in secondary metabolites, especially phytotoxins, due to the presence of milky or colored sap; the name of their family (spurge family) is derived from it [26].

In Saudi Arabia, there are nearly 40 species of the genus, but in the Jazan region, that located at the southwestern part of Saudi Arabia, there are nearly 15 species [27] that most of which are succulent with sharp spines, while others are neither succulent nor owing spines [28].

This study aims to evaluate the interrelations among four *Euphorbia* species at Jazan region by using two different categories: succulent with sharp spines; *E. ammak* (EA) and *E. fractiflexa* (EF) besides non-succulent without sharp spines; *E. granulata* (EG) and *E. hirta* (EH) upon morphology besides the modern one; chemotaxonomy.

## 2. Materials and methods

### 2.1. Collection of plant material and preparation

Four selected species of *Euphorbia*: *E. ammak, E. fractiflexa, E. granulata,* and *E. hirta* were collected from mountains, high latitudes, valleys, and inhabited in-situ areas respectively at Jazan region during Sept. 2022. They were kept and identified through the herbarium investigator for the College of Science, Jazan University, KSA 'JAZUH; Dr. Remesh Moochikkal. Coordinates from the maps of the Saudi Survey Authority (Table 1) (Figs 1 and 2).

The plant samples were analyzed to determine the morphological characters that were classified as qualitative, *i.e.*, the character presents or not, as well as quantitative, *i.e.*, the character can be measured. Micro-capillar ruler and a magnified lens were used in the analysis. After that, they were washed with dist. water 2–3 times, then dried at 55 °C till constant weight, finally cut into small pieces, preluding to grind as a powder sample for phytochemical analysis [29].

## 3. Estimation of phytochemical active constituents

Using the following techniques, the total active components of each of the following species; *E. ammak, E. fractiflexa, E. granulata,* and *E. hirta* were estimated separately.

### 3.1. Total phenolic content (TPC)

It was measured using the Folin-Ciocalteau technique [31,32]. 3 ml of 10% solution was mixed with 5 μl of plant extract and 0.8 ml of 7.5% sodium bicarbonate. The reaction solution was incubated at room temperature for 30 min. The mixture's absorbance was determined at 765 nm using a Milton Roy (Spectronic 1201) spectrophotometer. A measure of the TPC was mg of gallic acid equivalents (GAE)/g of extract.

Table 1. Coordination and location of *Euphorbia* species collected.

| species | Collecting Coordinates (Lat. N, long E.) | Locality |
|---|---|---|
| *E. ammak* | 17°20' N 43°08' E | Jabal Fayfa |
| *E. fractiflexa* | 17°08' N 43°05' E | Jabal Abadil |
| *E. granulata* | 17°11' N 43°10' E | Wadi Dahan, |
| *E. hirta* | 17°25' N 42°35' E | Wadi Baysh |

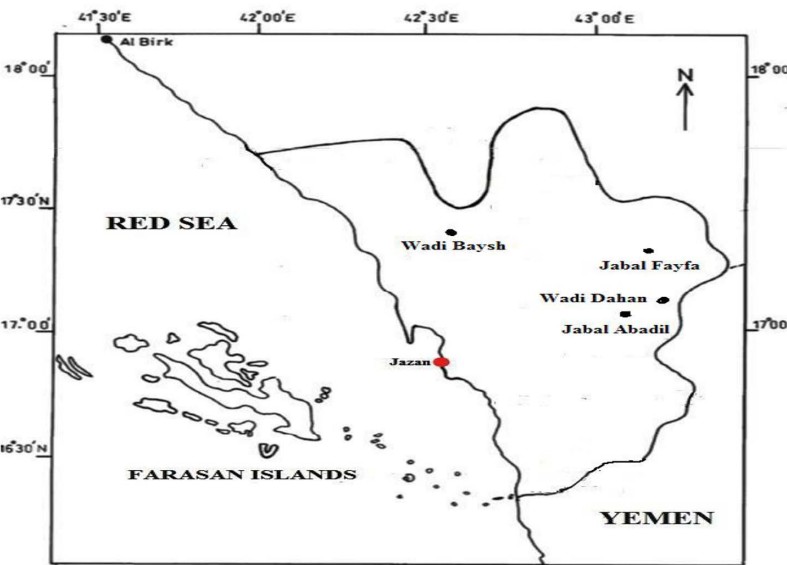

**Fig 1. Map of the studied area [30].**

### 3.2. Total flavonoid content (TFC)

The Chang *et al.* (2002) [3] technique was used to quantify TFC. In short, 3.9 ml of dist. water, 0.3 ml of 5% sodium nitrite solution, and 0.1 ml of extract were mixed. The mixture was left to react for 5 min before 0.3 ml of 10% aluminum chloride solution was added. After another 6 min of reaction, the mixture was treated with 2 mL of sodium hydroxide (1 M). Finally, each sample received 2.4 milliliters of dist. water. Using a Milton Roy (Spectronic 1201) spectrophotometer, the absorbance was measured at 510 nm against a sample blank that had not undergone any reaction. The extract's TFC is given as mg of quercetin equivalents (QE) per gram of extract.

### 3.3. Content of total alkaloids

Using the methodology outlined by Harbone (1973) [33], the total alkaloids of the aerial sections of *Euphorbia* plants were quantified. 200 ml of 20% acetic acid in ethyl alcohol and 3 g of plant powder were mixed in a 250 ml conical flask. The conical was then covered and allowed to stand for 6 h using a water bath. The plant extract was filtered and concentrated to 25% of its initial volume. To the concentrated extract, a concentrated $NH_4OH$ solution was added drop by drop until full precipitation. After letting the whole mixture settle, the precipitate was gathered, filtered, and weighed.

### 3.4. Total saponin content

In a 250 ml conical flask, 25 ml of 20% ethanol was mixed with about 50 mg of the plant powder. In short, the suspension was continuously stirred while heated to 55 °C for 4 h. After the mixture was filtered, 50 ml of 20% ethanol was used to remove the residue once more. The mixture of extracts was concentrated to 10 mL at 90°C. In order to create a diethyl ether layer, the concentrated extract was thoroughly agitated with 5 mL of diethyl ether in a separating funnel. The ether layer was then removed, allowing the aqueous layer to be recovered. There was another round of this cleansing. After adding 15 ml of n-butanol to the aqueous layer and shaking to create two layers, the butanol layer was collected 3 times. 10 ml of 5% aqueous sodium chloride was used to completely wash the mixed n-butanol extracts twice. The saponin content was determined by evaporating the residual solution in a water bath, drying it in an oven, and weighing it [24].

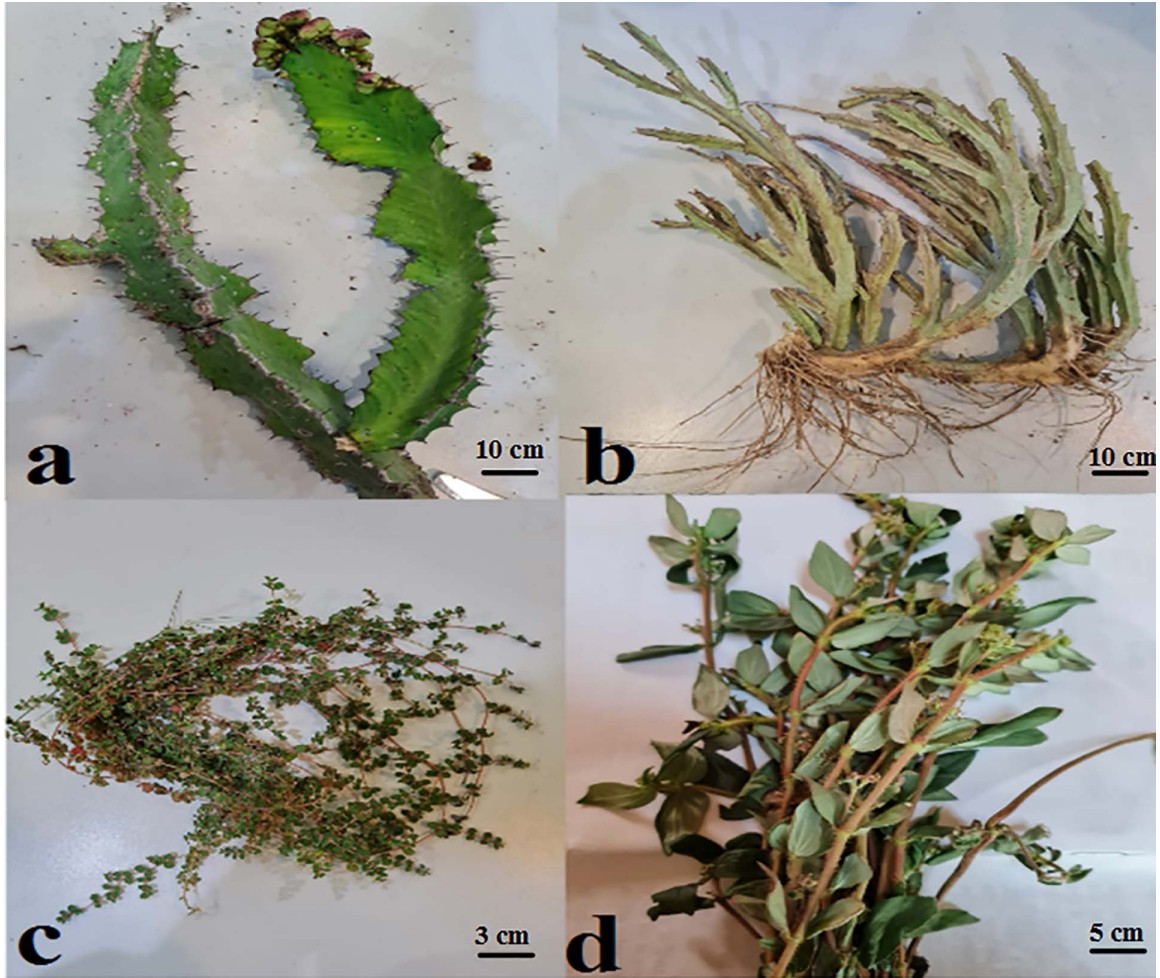

**Fig 2. The studied *Euphorbia* spp.; a. *E. ammak*, b. *E. fractiflexa*, c. *E. granulata*, d. *E. hirta*.**

### 3.5. Total tannin content

It was calculated using the copper acetate method, which relies on quantitative tannin precipitation with copper acetate solution, copper tannate ignition to copper oxide, and residual copper oxide measurement. Two grams of plant powder were extracted by agitating two separate volumes of 100 ml each of acetone and water (1:1) for 1 h in a mechanical shaker, followed by filtering. After transferring the mixed extract into a 250 ml volumetric flask, dist. water was used to adjust the volume. After measuring out each extract and placing it in a 500 ml beaker, 30 ml of a 15% aqueous copper acetate solution was added while stirring. The ashless filter paper was used to collect the precipitated copper tannate, which was then fired in a porcelain crucible (the crucibles had been ignited to a constant weight at the same temperature beforehand). The residue was mixed with a few drops of nitric acid and re-ignited to a consistent weight. The following correlation was used to compute the weight of copper oxide and the proportion of tannin: For every 1g of CuO, there was approximately 1.305 g of tannins [1].

### 4. HPLC analysis of phenolics and flavonoids

HPLC equipment (Agilent Series 1100; Agilent, USA) comprising an auto-sampling injector, solvent degasser, two LC pumps (series 1100), ChemStation software, and a UV/Vis detector (set at 250 nm for phenolic acids and 360 nm for

flavonoids) was used to analyze phenolic and flavonoid compounds. A C18 column (125 mm × 4.60 mm, 5 µm particle size) was used for the study. Using a gradient mobile phase of solvents A and B (A = methanol; B = acetic acid in water (1:25)), phenolic acids were isolated. For the first three minutes of the gradient procedure, the concentration was maintained at 100% B. This was followed by 50% eluent A for the next 5 minutes, after which the concentration of A was increased to 80% for the next 2 minutes and then reduced to 50% again for the following 5 min a detection wavelength of 250 nm. Flavonoids were separated by employing a mobile phase of solvents C and D [C = acetonitrile; D = 0.2% (v/v) aqueous formic acid] with an isocratic elution (70:30) program. The solvent flow rate was 1 ml/min, and separation was performed at 25°C. The injection volumes were 25 µl [34,35].

## 5. Scoring data with cluster analysis

To create a cluster analysis for *Euphorbia* spp., the morphological and phytochemical parameters were scored as a binary matrix, with (0) denoting absence and (1) denoting presence. Distances were computed using a Gower coefficient modification, and a similarity matrix was created using Pclass. UPGMA was used for hierarchical nest clustering that was sequentially agglomerative by using 'SAHN' sub-program of NTS-YS-PC software [36].

## 6. Statistical analysis

Duncan's range test was used to determine whether the results of quantitative phytochemical compounds were significant or not. It was used to differentiate among plant species in the presence of a, b, and c codes by conducting SPSS software (version 22) for Windows [37].

## 7. Results and discussion

### 7.1. Morphological analysis

Morphological characters included different aspects of plant species. They indicated the plant habit, life span, and growth. Moreover, the reproductive plant parts were present in this investigation (fruit and inflorescence). Each plant part occupied a different percent for presentation from total morphological characters: stem, 31.82%, flowers, 27.27%, fruit, 22.73%, leaf, 9.09% and root, 4.54%. The characters were divided as qualitative, like color, branching, and others, besides quantitative, like fruit dimensions and pedicle length, etc. The macro parameter, qualitative character, possessed 81.82% while the micro parameter, quantitative character, possessed 18.18%. The similarity parameter is considered the common character among the studied plant species. It was 9.09% while the dissimilarity parameter was 90.91%. Unique parameter is the character that is only restricted to one plant species and not present in the others. It was 27.27% distributed in only three species: *E. fractiflexa, E. granulata*, and *E. hirta.* Furthermore, *E. granulata* occupied the highest value; 50% on the contrary, *E. hirta* had the lowest one, 16.67% (Table 2).

### 7.2. Phytochemical analysis

The studied plant species exhibited different values with phytochemical data in all classes of secondary metabolites. *E. granulata* was recognized in this investigation because it occupied the top in all classes. On one hand, it had the highest percent in phenolic, flavonoid, alkaloid, saponin, and tannin content with 350.29 mg/g, 86.57 mg/g, 7.48%, 6.87% and 10.5% respectively. On the other hand, two plant species occupied the lowest level in different secondary metabolite classes.

   *E. hirta* was the first one owing the lowest values in three classes: flavonoids (8.04 mg/g), alkaloids (3.46%), and tannins (3.09%), while *E. fractiflexa* was the second one owing lowest values in two classes: phenolics (45.69 mg/g) and saponins (3.44%) (Tables 3 and 4).

   HPLC analyzes data for phenolic and flavonoid constituents of the four tested *Euphorbia* species, which were listed in Table 3 and Figs 3–5. Eleven phenolic (benzoic acid, catechol, caffeic acid, cinnamic acid, ferulic acid, gallic acid, pyrogallol,

**Table 2. Morphological characters of studied plant species.**

| | Morphological character | *E. ammak* | *E. fractiflexa* | *E. granulata* | *E. hirta* |
|---|---|---|---|---|---|
| 1. Stems | Branching | Many | Many | Many | Sparingly |
| | Surface | Spineless | Spiny | Spineless | Spineless |
| | Color | Light green | Light green | Dark green | Dark green |
| | Succulence | Succulent | Succulent | Non- Succulent | Non-Succulent |
| | Length | Very tall | Tall | Very short | Short |
| | Circumference | Triangle | Triangle | Rounded | Rounded |
| | Erection | Erect | Decumbent | Decumbent | Erect |
| 2. Inflorescences | Color | Yellow | Yellow | White | Yellow |
| | Width | 7 ± 0.05 mm | 6 ± 1.05 mm | 1 ± 1.75 mm | < 1 mm |
| | Type | Cyme | Cyme | Raceme | Cyme |
| | Position | Terminal | Terminal | Axillary | Axillary |
| | Peduncle | < 1 cm | < 1 cm | < 1 cm | > 1 cm |
| 3. Fruits | Color | Reddish brown | Reddish brown | Gray | Gray |
| | Type | Capsule | Capsule | Capsule | Capsule |
| | Dimensions | 8.5 x 13 mm | 9 x 14 mm | 1.5 x 1.5 mm | 1.5 x 1.5 mm |
| | Trichomes | Glabrous | Glabrous | Pubescent | Pubescent |
| | Shape | Triangular, acute | Trigonous, rounded | Angular lobes | Angular lobes |
| | Pedicel length | 2.5 ± 1.59 mm | 3 ± 1.78 mm | 0.5 ± 1.59 mm | 1.2 ± 1.48 mm |
| 4. Roots | Root length | not deep | not deep | not deep | not deep |
| 5. Leaves | Lamina | Leafless | Leafless | Compound | Simple |
| | Leaf trichomes | Absent | Absent | Present | Absent |
| 6. Life span | Duration | Perennial | Perennial | Annual | Annual |
| 7. Habit | Life form | Shrubs | Shrubs | Herb | herb |

**Table 3. Total phenolics and flavonoids contents.**

| Sample Code | Total Phenolics mg/g | | | | | Total Flavonoids (mg/g) | | | | |
|---|---|---|---|---|---|---|---|---|---|---|
| | 1st rep. | 2nd rep. | 3rd rep. | Mean | S.D. (±) | 1st rep. | 2nd rep. | 3rd rep. | Mean | S.D. (±) |
| *E. ammak* | 136.56 | 130.41 | 125.92 | 130.96[ab] | 5.34 | 83.20 | 78.35 | 76.56 | 79.37[a] | 3.44 |
| *E. fractiflexa* | 39.65 | 51.29 | 46.13 | 45.69[c] | 5.83 | 33.02 | 31.79 | 28.14 | 30.98[b] | 2.54 |
| *E. granulata* | 371.25 | 314.69 | 364.93 | 350.29[a] | 30.99 | 86.57 | 88.23 | 84.91 | 86.57[a] | 1.66 |
| *E. hirta* | 73.84 | 71.39 | 80.27 | 75.17[c] | 4.59 | 6.92 | 8.17 | 9.04 | 8.04[c] | 1.07 |

*The assay was performed in triplicate.

syringenic acid, *p*-coumaric acid, salicylic acid, and chlorogenic acid) and seven flavonoid (apigenin, kaempferol, luteolin, myricetin, naringin, quercetin, and rutin) compounds were identified from the four species with different relative percent.

Caffeic acid, gallic acid, and syringic acid were the mutual phenolic acids among the four species, while apigenin and glycoside naringin were the mutual flavonoids.

*E. ammak* showed a high percentage of caffeic acid, gallic acid, and apigenin, while *E. hirta* showed a high percentage of syringenic acid, and *E. fractiflexa* showed a high percentage of apigenin among the four species. Benzoic acid, cinnamic acid, salicylic acid, and kaempferol were the mutual compounds between *E. fractiflexa*, *E. granulata*, and *E. hirta* species, while *p*-coumaric acid, luteolin, and quercetin were the mutual compounds between *E. ammak*, *E. granulata*, and *E. hirta* species.

**Table 4. Total content of alkaloids, saponin, and tannins.**

| Sample ID | Total alkaloids content | | | Total Saponin contents | | | Total tannins contents | | | |
|---|---|---|---|---|---|---|---|---|---|---|
| | SW (g) | PW (g) | % | SW (g) | PW (g) | % | SW (g) | PW (g) | TW (g) | % |
| *E. ammak* | 2.504 | 0.1169 | 4.66[ab] | 2.05 | 0.0784 | 3.80[ab] | 2.00 | 0.09609 | 0.1254 | 6.27[ab] |
| *E. fractiflexa* | 2.513 | 0.1656 | 6.59[a] | 2.03 | 0.0699 | 3.44[ab] | 2.03 | 0.1314 | 0.1715 | 8.57[ab] |
| *E. granulata* | 2.523 | 0.1886 | 7.48[a] | 2.07 | 0.1423 | 6.87[a] | 2.07 | 0.16122 | 0.2104 | 10.5[a] |
| *E. hirta* | 2.519 | 0.0872 | 3.46[ab] | 2.06 | 0.1102 | 5.37[a] | 2.06 | 0.05118 | 0.0668 | 3.09[c] |

SW = weight of sample; PW = weight of precipitate; TW = weight of tannins.

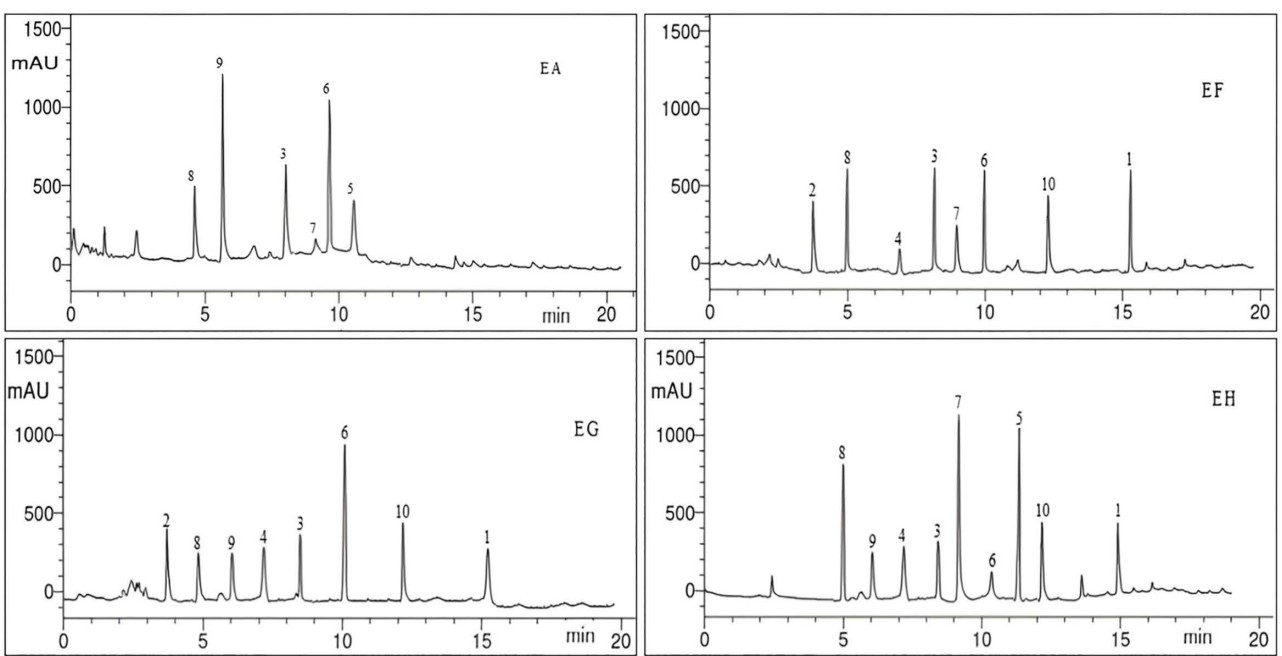

**Fig 3. HPLC Chromatogram of the phenolic components of *E. ammak*, *E. fractiflexa*, *E. granulata*, and *E. hirta* methanolic extracts Peaks respectively: 1, benzoic acid; 2, catechol; 3, caffeic acid; 4, cinnamic acid, 5, ferulic acid; 6, gallic acid; 7, pyrogallol; 8, syringenic acid; 9, *p*-coumaric acid; 10, salicylic acid.**

Pyrogallol was the mutual phenolic between *E. ammak*, *E. fractiflexa*, and *E. hirta*, while ferulic acid was mutual between *E. ammak* and *E. hirta* only.

Catechol was the mutual phenolic between *E. fractiflexa* and *E. granulata*. Chlorogenic acid and rutin were identified only from the *E. ammak* species. Myricetin flavonoid was identified only in *E. fractiflexa* species.

*E. ammak* had the highest values for all identified phytochemicals: *p*-coumaric acid (21.56 µg/g) and apigenin (21.76 µg/g), while the lowest value was obtained by *E. hirta* at gallic acid (0.98 µg/g).

The similarity parameter among identified phytochemicals was 44.44%. Negative unique parameter (NUP) is considered the parameter that is absent in a specific plant species but present in others. It occupied 33.33% and was equally in the two species: *E. ammak* and *E. fractiflexa*.

Nevertheless, there was another type of parameter, a positive unique parameter (PUP). It is considered that the parameter is present only in one species, not others. It was 11.11% and located only at *E. ammak* in the presence of chlorogenic acid and rutin (Table 5) (Figs 3 and 4). The chemical structure of the identified phenolic and flavonoids compounds was illustrated in Fig 6.

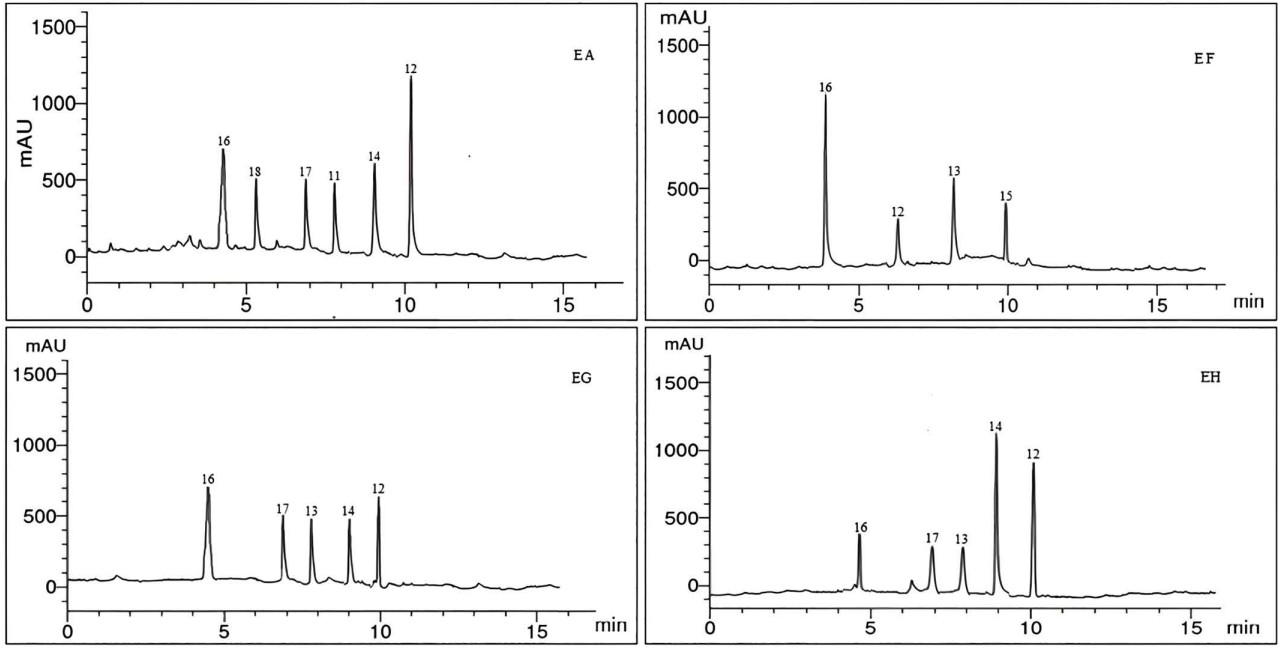

**Fig 4. HPLC Chromatogram of the flavonoid components of *E. ammak*, *E. fractiflexa*, *E. granulata*, and *E. hirta* methanolic extracts respectively:** 11, chlorogenic acid; 12, apigenin; 13, kaempferol; 14, luteolin; 15, myricetin; 16, naringin; 17, quercetin; 18, rutin.

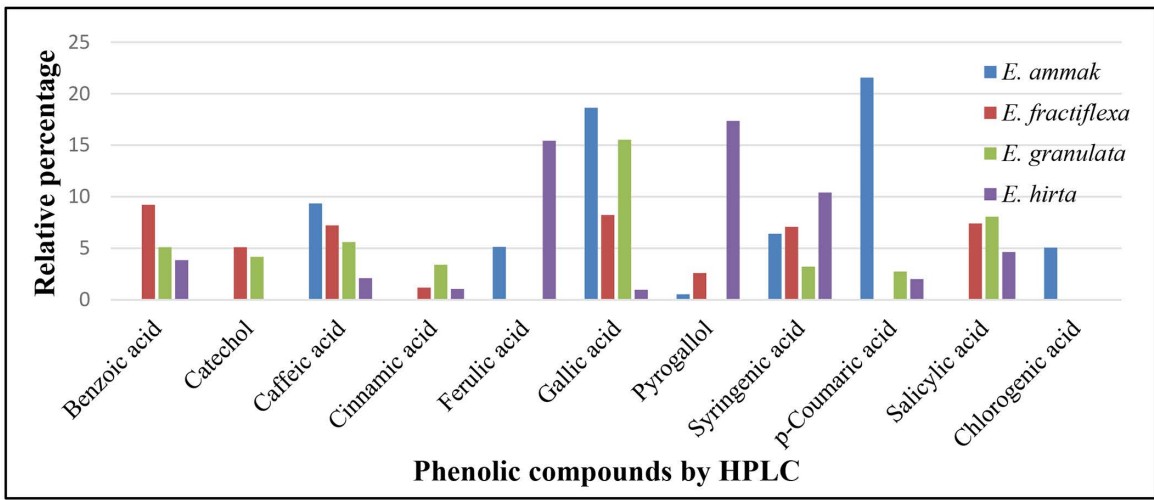

**Fig 5. Relative percent of phenolic compounds of *Euphorbia* spp. using HPLC.**

## 7.3. Scoring analysis

A total of 45 traits for both morphological and phytochemicals were scored as a binary matrix to differentiate among the studied plant species. The binary matrix is denoted as (0) absent and (1) present. For morphological characters, the common character was represented with (1) while the unique or less common was (0). On the other hand, total active constituent content was represented with (1) donation for the highest values, while (0) donation was the lowest. Moreover,

**Table 5. Relative percent of phenol and flavonoid components of *Euphorbia* species methanolic extracts screened by HPLC.**

| # | Compound Name | Concentration µg/g | | | |
|---|---|---|---|---|---|
| | | *E. ammak* | *E. fractiflexa* | *E. granulata* | *E. hirta* |
| Phenolic compounds | | | | | |
| 1 | Benzoic acid | – | 9.22[a] | 5.11[ab] | 3.85[b] |
| 2 | Catechol | – | 5.1 | 4.18 | |
| 3 | Caffeic acid | 9.36[a] | 7.22[a] | 5.6a[b] | 2.1[b] |
| 4 | Cinnamic acid | – | 1.19[ab] | 3.41[a] | 1.06[ab] |
| 5 | Ferulic acid | 5.14[b] | – | – | 15.44[a] |
| 6 | Gallic acid | 18.64[a] | 8.23[b] | 15.54[a] | 0.98[c] |
| 7 | Pyrogallol | 0.54[c] | 2.61[c] | – | 17.37[a] |
| 8 | Syringenic acid | 6.41[a] | 7.08[a] | 3.22[c] | 10.41[a] |
| 9 | *p*-Coumaric acid | 21.56[a] | – | 2.74[c] | 2.02[c] |
| 10 | Salicylic acid | – | 7.41[a] | 8.07[a] | 4.63[ab] |
| 11 | Chlorogenic acid | 5.07 | – | – | – |
| Flavonoid compounds | | | | | |
| 12 | Apigenin | 21.76[a] | 5.2[c] | 9.85[b] | 12.74[b] |
| 13 | Kaempferol | – | 8.19[a] | 4.24[ab] | 5.18[ab] |
| 14 | Luteolin | 10.41[a] | – | 6.32[b] | 15.26[a] |
| 15 | Myricetin | – | 6.78 | – | – |
| 16 | Naringin | 8.95[ab] | 15.36[a] | 8.57[ab] | 4.76[b] |
| 17 | Quercetin | 6.29[a] | – | 5.44[a] | 6.41[a] |
| 18 | Rutin | 6.41 | – | – | – |

The same letter refers to non-different.

the absence of phytochemical compounds from HPLC is denoted with (0); meanwhile, all present phytochemicals with different concentrations were denoted with (1).

The similarity matrix was obtained to indicate that the highest score was between *E. granulata* and *E. hirta* (0.69); however, the lowest one was between *E. ammak* and *E. granulata*.

The phenogram showed the interrelations among the studied plant species. The taxonomic deviation for them was at 1.06 taxonomic units, where two groups appeared. The first group was at 1.90, included *E. ammak* and *E. fractiflexa*, while the other group was at 1.52, and included the rest of the plant species (Table 6) (Fig 7).

Taxonomy of the genus *Euphorbia* faces discerned challenges and implications owing to insufficient taxon collection, homoplasious and obscure morphological traits, in addition to different biogeography. Other taxonomic tool, like chemotaxonomy, is integrated to enhance phylogenetic interrelationships and evolution among them [38,39].

Chemotaxonomy is regarded as an efficient and vital tool for plant taxonomy. Combining modern and traditional classifications can promote the accurate taxonomic methodology [40].

Although the studied plant species were divided into two groups, obviously not based only on morphological characters but confirmed with phytochemicals, *E. granulata* was present as a transition species among the studied ones. It had the majority of identified phytochemical contents, though it had milky sap. It had the majority of identified phytochemicals by HPLC. In addition to that, *E. ammak* (succulent with milky sap) also had other separated phytochemicals not present in *E. granulata* and vice versa. Although a distant variation in morphological characters between them, *i.e.*, *E. ammak* is tree, succulent with milky sap; on the contrary, *E. granulata* is decumbent, ephemeral, non-succulent without milky sap, this situation confirms that there is a link between them, leading both to be in

**Fig 6. The chemical structures of the identified phenolic and flavonoids compounds.**

**Table 6. Similarity matrix among *Euphorbia* spp.**

|  | *E. ammak* | *E. fractiflexa* | *E. granulata* | *E. hirta* |
|---|---|---|---|---|
| *E. ammak* | 1.00 | 0.62 | 0.38 | 0.44 |
| *E. fractiflexa* | 0.62 | 1.00 | 0.44 | 0.42 |
| *E. granulata* | 0.38 | 0.44 | 1.00 | 0.69 |
| *E. hirta* | 0.44 | 0.42 | 0.69 | 1.00 |

one family. This distant variation may refer to the nature of habitats. Adaptation of plant species to their habitat may prove to be more variable than other plant species belonging to the same family. Phytochemicals are reported to help the plant resist and withstand its habitat [41,42].

This systematic investigation revolves around the importance of phytochemical secondary metabolites in plant taxonomy aspects as well as the link between these compounds and specific environmental locality with different responses.

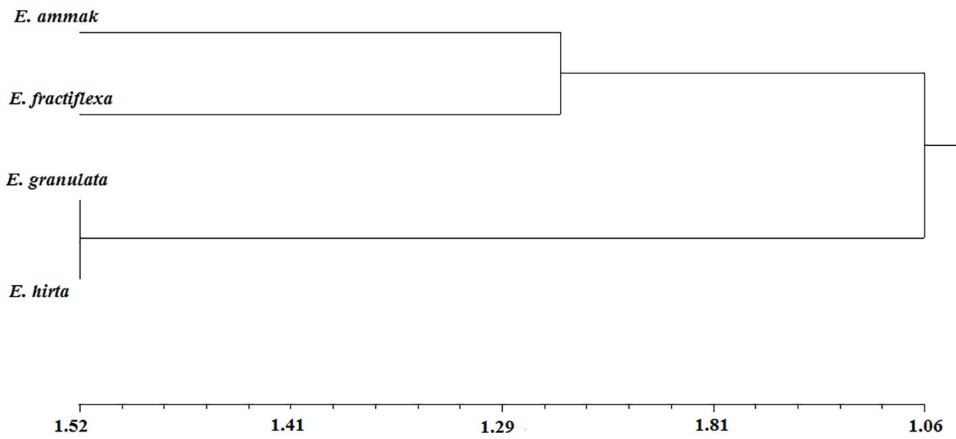

**Fig 7. Phenogram for the studied *Euphorbia* spp.**

World ecological distribution and various morphological variability of *Euphorbia* spp. induce different categories of secondary metabolites that attract most researchers in taxonomical, pharmacological, medicinal, and physiological studies since modern and ancient times.

Continuous phytochemical screening for different species belonging to the same genus at different localities all over the world encourages collecting data serving humanity in different fields of life.

The presence of phytochemicals with high concentrations in the form of secondary metabolites encourages researchers to use this family in pharmacological and medicinal studies. Even the same plant habitat and behavior exposed different concentrations and classes of secondary metabolites that provide more varied phytochemical classes when located in different or harsh environments.

## 8. Conclusion

The study of phytochemicals should be done while taking into consideration the habitat of plant species, especially if the plant species belongs to a remarkable family with rich secondary metabolites. Not all milky sap is rich in phytochemicals, but it may contain other, more toxic compounds in large quantities. Chemotaxonomy should be undertaken side by side with traditional taxonomy to give the full picture of interrelations among plant species. The current characterization of the secondary metabolites is a beneficial tool for discrimination between the four closely related *Euphorbia* species, which supports their morphological variation.

## Author contributions

**Conceptualization:** Yehia Hazzazi, Mari Sumayli.

**Data curation:** Abeer Al-Andal.

**Formal analysis:** Uzma Hanif.

**Funding acquisition:** Yehia Hazzazi, Mari Sumayli.

**Investigation:** Asmaa Khamis.

**Methodology:** A. El-Shabasy, Ahmed A. M. Abdelgawad.

**Software:** Emad Abada.

**Validation:** Sameh R. Elgogary.

**Visualization:** Taha A. I. El-Bassossy.

**Writing – original draft:** Ahmed A. M. Abdelgawad.

**Writing – review & editing:** A. El-Shabasy.

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
