## [Decision Letter · Decision Letter 0]

3 Jul 2025

Dear Dr. El-Shabasy,

Please submit your revised manuscript before Aug 17 2025 11:59PM. If you will need more time than this to complete your revisions, please reply to this message or contact the journal office at plosone@plos.org . A rebuttal letter that responds to each point raised by the academic editor and reviewer(s). You should upload this letter as a separate file labeled 'Response to Reviewers'.A marked-up copy of your manuscript that highlights changes made to the original version. You should upload this as a separate file labeled 'Revised Manuscript with Track Changes'.An unmarked version of your revised paper without tracked changes. You should upload this as a separate file labeled 'Manuscript'.

We look forward to receiving your revised manuscript.

Kind regards,

Waqas Khan Kayani, PhD

Academic Editor

PLOS ONE

3. Please ensure that you refer to Figure 3, 4, 5, 6 in your text as, if accepted, production will need this reference to link the reader to the figure.

Additional Editor Comments (if provided):

Reviewers' comments:

Reviewer's Responses to Questions

**Comments to the Author**

1. Is the manuscript technically sound, and do the data support the conclusions?

Reviewer #1: Yes

Reviewer #2: Yes

2. Has the statistical analysis been performed appropriately and rigorously?

Reviewer #1: Yes

Reviewer #2: Yes

3. Have the authors made all data underlying the findings in their manuscript fully available?

Reviewer #1: Yes

Reviewer #2: Yes

4. Is the manuscript presented in an intelligible fashion and written in standard English?

Reviewer #1: Yes

Reviewer #2: No

Reviewer #1: The data support the conclusion, the statistical analysis has been performed appropriately and rigorously, the authors made all data underlying the findings in their manuscript fully available , the manuscript presented in an intelligible fashion and written in standard English, The abstract must be include more results ,the materials should be abbreviation ,in abstract ,you must write full scientific name ,and in rest text you can abbreviation the genus name ,in results don't write the scientific name in wrong methods ,all comments was written in the text

Reviewer #2: Reviewer Report

Journal: POLS Journal

Manuscript Title: Chemotaxonomic and Morphological Delimitation of Selected Euphorbia Species from Jazan Region, Saudi Arabia

Reviewer: Dr. Amir Shahbaz

Date: June 25, 2025

General Assessment

The manuscript presents an integrative study on the classification of four Euphorbia species using both traditional morphological and modern chemotaxonomic approaches. The authors employed quantitative phytochemical analyses (including HPLC profiling of phenolics and flavonoids) and morphological scoring to assess interspecific relationships. The research topic is timely and relevant, particularly for floristic, taxonomic, and phytochemical studies within arid regions.

Major Revisions Needed

1. Language and Grammar: The manuscript needs thorough English language editing by a native or fluent scientific editor.

2. . Scientific Terminology: Replace vague or incorrect terms with standardized scientific terminology.

3. Data Presentation: Improve table formatting and figure resolution; ensure clarity in HPLC chromatograms.

4. Introduction and Background: Improve paragraph transitions and clearly state the rationale for species selection.

5. Conclusion: Revise to avoid generalizations and ensure scientific clarity.

Minor Points

Ensure species names are italicized throughout.

Correct and standardize reference formatting.

Improve phenogram resolution and labeling.

Clarify methodology in scoring matrix.

Recommendation

Major Revision Required - The manuscript offers valuable taxonomic insights but needs substantial revision in language, terminology, and presentation before it can be considered for publication.

**Do you want your identity to be public for this peer review?** For information about this choice, including consent withdrawal, please see our Privacy Policy

Reviewer #1: **Yes: ** Professor Dr.Rana Hashim Aloush

Reviewer #2: No

---

## [Author Response · Author response to Decision Letter 1]

18 Aug 2025

Dear editor and reviewers

I follow all your instructions at revised manuscript

---

## [Decision Letter · Decision Letter 1]

14 Oct 2025

Morphological and chemotaxonomical characterization of some species of the genus Euphorbia L. in Jazan region, KSA

PONE-D-25-25839R1

Dear Dr. Ahmed El-Shabasy,

We’re pleased to inform you that your manuscript has been judged scientifically suitable for publication and will be formally accepted for publication once it meets all outstanding technical requirements.

Kind regards,

Waqas Khan Kayani, PhD

Academic Editor

PLOS ONE

Additional Editor Comments (optional):

Reviewers' comments:

Reviewer's Responses to Questions

**Comments to the Author**

Reviewer #2: All comments have been addressed

2. Is the manuscript technically sound, and do the data support the conclusions?

Reviewer #2: Yes

3. Has the statistical analysis been performed appropriately and rigorously?

Reviewer #2: Yes

4. Have the authors made all data underlying the findings in their manuscript fully available?

Reviewer #2: Yes

5. Is the manuscript presented in an intelligible fashion and written in standard English?

Reviewer #2: Yes

Reviewer #2: In my initial review, I recommended major revisions to improve the clarity, methodological rigor, and overall presentation of the manuscript. The authors have carefully addressed all the raised concerns and incorporated the suggested changes in the revised version.

The revised manuscript is now significantly improved in terms of scientific content, organization, and readability. The responses provided by the authors were satisfactory, and the additional analyses/clarifications strengthened the overall quality of the work.

**Do you want your identity to be public for this peer review?** For information about this choice, including consent withdrawal, please see our Privacy Policy

Reviewer #2: No

---

## [Editor Report · Acceptance letter]

PONE-D-25-25839R1

PLOS ONE

Dear Dr. El-Shabasy,

I'm pleased to inform you that your manuscript has been deemed suitable for publication in PLOS ONE. Congratulations! Your manuscript is now being handed over to our production team.

Kind regards,

on behalf of

Dr. Waqas Khan Kayani

Academic Editor

PLOS ONE